# HCV Infection and Liver Cirrhosis Are Associated with a Less-Favorable Serum Cholesteryl Ester Profile Which Improves through the Successful Treatment of HCV

**DOI:** 10.3390/biomedicines10123152

**Published:** 2022-12-06

**Authors:** Kilian Weigand, Georg Peschel, Jonathan Grimm, Martina Müller, Marcus Höring, Sabrina Krautbauer, Gerhard Liebisch, Christa Buechler

**Affiliations:** 1Department of Internal Medicine I, Gastroenterology, Hepatology, Endocrinology, Rheumatology and Infectious Diseases, University Hospital Regensburg, 93053 Regensburg, Germany; 2Department of Gastroenterology, Gemeinschaftsklinikum Mittelrhein, 56073 Koblenz, Germany; 3Department of Internal Medicine, Klinikum Fürstenfeldbruck, 82256 Fürstenfeldbruck, Germany; 4Institute of Clinical Chemistry and Laboratory Medicine, University Hospital Regensburg, 93053 Regensburg, Germany

**Keywords:** free cholesterol, cholesteryl ester, fibrosis-4 score, direct-acting antivirals, hepatitis C, liver cirrhosis

## Abstract

**Background**: Infection with hepatitis C virus (HCV) lowers serum cholesterol levels, which rapidly recover during therapy with direct-acting antivirals (DAAs). Serum cholesterol is also reduced in patients with liver cirrhosis. Studies investigating serum cholesterol in patients with chronic liver diseases are generally based on enzymatic assays providing total cholesterol levels. Hence, these studies do not account for the individual cholesteryl ester (CE) species, which have different properties according to acyl chain length and desaturation. **Methods**: Free cholesterol (FC) and 15 CE species were quantified by flow injection analysis high-resolution Fourier Transform mass spectrometry (FIA-FTMS) in the serum of 178 patients with chronic HCV before therapy and during treatment with DAAs. **Results**: Serum CEs were low in HCV patients with liver cirrhosis and, compared to patients without cirrhosis, proportions of CE 16:0 and 16:1 were higher whereas % CE 20:4 and 20:5 were reduced. FC levels were unchanged, and the CE/FC ratio was consequently low in cirrhosis. FC and CEs did not correlate with viral load. Four CE species were reduced in genotype 3 compared to genotype 1-infected patients. During DAA therapy, 9 of the 15 measured CE species, and the CE/FC ratio, increased. Relative to total CE levels, % CE 16:0 declined and % CE 18:3 was higher at therapy end. At this time, % CE 14:0, 16:0 and 16:1 were higher and % CE 20:4 and 22:6 were lower in the cirrhosis than the non-cirrhosis patients. Viral genotype associated changes of CEs disappeared at therapy end. **Conclusions**: The serum CE composition differs between patients with and without liver cirrhosis, and changes through the efficient elimination of HCV. Overall, HCV infection and cirrhosis are associated with a higher proportion of CE species with a lower number of carbon atoms and double bonds, reflecting a less-favorable CE profile.

## 1. Introduction

Chronic infection with hepatitis C virus (HCV) is a common cause of liver fibrosis, which may progress to cirrhosis [1]. Metabolic diseases such as liver steatosis or diabetes mellitus occur more often in association with chronic HCV infection. Fatty liver and diabetes can be direct effects of viral infection or a secondary event of insulin resistance due to viral infection. HCV patients may also suffer from the metabolic syndrome, which affects about 30% of the normal population. Thus, it is difficult to distinguish between these different causes of liver steatosis and insulin resistance. Hypertension and obesity are indicators of metabolic diseases, and HCV genotype 3 infection in patients with normal body weight is suggestive of viral fatty liver [2,3]. Effective eradication of HCV improves insulin resistance and liver steatosis related to viral infection but cannot resolve fatty liver disease and impaired insulin response as a component of the metabolic syndrome. Metabolic liver steatosis and insulin resistance contribute to the progression of liver fibrosis and patients are at a higher risk for secondary complications such as cardiovascular diseases [3].

Direct-acting antivirals (DAAs) have emerged as highly efficient therapeutics, which eliminate HCV within a short time, and a sustained virologic response (SVR) of up to 100% can be achieved [4,5,6]. The model for end-stage liver disease (MELD) score is widely used for the assessment of liver disease severity [7]. Most of the patients followed for up to four years after treatment with DAAs had only marginal improvements of the MELD score, showing that liver dysfunction persists [8,9]. It was also reported that about 30% of patients had an improved MELD score ≥ 3 at sustained virologic response rates at off-treatment week 12 when treated with sofosbuvir/velpatasvir plus ribavirin [10]. Notably, DAA therapy is similarly effective in HCV patients with and without liver cirrhosis [10,11]. 

The diagnosis of liver cirrhosis in clinical practice is based on physical examination, laboratory parameters and ultrasound. Liver biopsy is usually not necessary to confirm liver cirrhosis [12]. The histological evaluation of a liver biopsy is regarded as gold standard for the staging of fibrosis. However, liver biopsy has several limitations and potential adverse events [12,13]. Therefore, non-invasive methods have been established for the assessment of liver fibrosis. In general, these methods display a high diagnostic accuracy for the exclusion of liver fibrosis and for the diagnosis of advanced liver fibrosis [13]. The fibrosis-4 (FIB-4) score is calculated from age, aspartate aminotransferase (AST; U/L), alanine aminotransferase (ALT; U/L) and platelet count (×10^9^/L) [14]. A high FIB-4 score was validated as an accurate marker of advanced liver fibrosis in HCV patients [15]. 

Chronic HCV infection is associated with low levels of low-density lipoprotein (LDL) and serum cholesterol. Most, if not all, studies on DAA treatment so far have consistently shown that serum total cholesterol and LDL levels are induced early after the start of treatment [8,16,17,18]. Blood cholesterol exists as free cholesterol (FC) and different cholesteryl ester (CE) species [19,20]. Whether FC and the different CE species equally increase during therapy has not been studied in great detail so far. 

CE fatty acid composition is of importance for its biological functions. In human plasma, most of the CE species are produced by the activity of lecithin-cholesterol acyltransferase (LCAT) [21]. By esterifying FC derived from peripheral tissues, LCAT promotes reverse cholesterol transport, and thus, has an essential role in whole body cholesterol homeostasis [22]. LCAT forms CE 20:4, 22:5 and 22:6, whereas CE 16:0, 18:1 and 18:3 are derived from liver acyl CoA: cholesterol acyltransferase (ACAT) [23,24,25]. ACAT2 is expressed in hepatocytes, and ACAT2-deficient mice had low serum cholesterol and were protected from atherosclerosis and hepatic lipid accumulation [26]. Further studies showed that ACAT2-derived CE species are predominantly atherogenic blood lipids [27]. LCAT overexpression resulted in higher HDL levels and prevented the development of diet-induced atherosclerosis [24]. The higher unsaturation of LDL-carried CEs was supposed to have beneficial cardiovascular effects [28]. LCAT is expressed in the liver and its activity is decreased in patients with liver cirrhosis [29]. Whether ACAT2 activity is also modified in the cirrhotic liver has not been studied as far as we know. 

HCV infection induces the hepatic ACAT2 mRNA expression of patients, and upregulation is more prominent in genotype 3- compared to genotype 1-infected patients [30]. CEs of hepatocytes are used for the production of infectious HCV particles, and this is impaired by the inhibition of ACAT [30]. LCAT activity was higher in HCV patients who did not achieve SVR in comparison to those with efficient clearance of the virus, suggesting that LCAT activity increases by HCV infection [31]. 

Patients with liver cirrhosis have low serum cholesterol, and LDL as well as high-density lipoprotein (HDL) are reduced [32,33,34]. The cholesterol esterification fraction, which was defined as level of esterified cholesterol vs. total cholesterol, is closely related to liver function [35]. In a cohort of patients with mixed disease aetiologies, and a median MELD score of 12, a low cholesterol esterification ratio was a good predictor of mortality [35]. A separate study calculated the CE/FC ratio but could not identify a difference between patients with predominantly alcoholic liver cirrhosis and liver-healthy controls [36].

Liver cirrhosis as well as HCV infection are characterized by low serum cholesterol [16,30,31,32]. Whether FC and the different CE species are all similarly reduced has not been analyzed in great detail so far. FC has unique properties and high levels are cytotoxic [37]. An association of serum FC with bilirubin has been described in patients with alcoholic liver cirrhosis [38]. Notably, serum CE levels were not associated with markers of liver cirrhosis in this cohort [38].

The effective therapy of HCV elevates serum LDL and cholesterol levels, which are risk factors for cardiovascular diseases [39,40]. The elimination of HCV is, however, associated with a lower risk for cardiovascular diseases [3]. Not all of the CE species seem to contribute to an increased risk, and a higher polyunsaturated-to-saturated CE ratio protected from atherosclerosis [39]. Among the lipids with a strong predictive value for cardiovascular diseases were CE species with a low carbon number and a low double-bond content [41]. 

The main aim of the present analysis was to study the effect of DAA therapy on serum CE composition. It was also analyzed whether the CE profile differs between HCV patients with and without liver cirrhosis. Therefore, FC and 15 CE species were measured in the serum of HCV patients before therapy, at 4 weeks after treatment start—the time point where LDL levels have been recovered [8,18]—and at therapy end.

## 2. Materials and Methods

### 2.1. Study Cohort

Patients’ sera were collected at the Department of Internal Medicine I (University Hospital of Regensburg) from October 2014 to September 2019 [42]. All of the 178 patients asked to participate in the study were suited for therapy with DAAs in agreement with HCV treatment guidelines [4] and all of them finished the study. None of the patients had been treated for HCV before. The patients were older than 18 years and were not coinfected with HBV or HIV. 

All patients had chronic HCV infection and were treated with one of the following regimens: sofosbuvir/daclatasvir; sofosbuvir/daclatasvir/ribavirin; sofosbuvir/ledipasvir; sofosbuvir/ledipasvir/ribavirin; sofosbuvir/velpatasvir; sofosbuvir/ribavirin and dasabuvir/ombitasvir/paritaprevir/ritonavir [4]. 

Very few patients were using statins, and due to the potential interaction of DAA therapy with statins [43], the cholesterol-lowering drugs were paused during therapy. 

Cirrhosis diagnosis by ultrasound was established based on nodular liver surface, small liver size and heterogeneous liver parenchyma [44]. Cut-off values used for the FIB-4 scoring were: >3.25: advanced fibrosis, <1.3: no fibrosis for patients younger than 65 years and <2: no fibrosis for patients older than 65 years [45]. 

Laboratory values were obtained from the Institute of Clinical Chemistry and Laboratory Medicine (University Hospital Regensburg). Individual laboratory values of ≥95% of the patients were available. Laboratory parameters of this study group have been published in an open access journal [46].

### 2.2. Measurement of FC and CEs

FC and CE were analyzed by flow injection analysis high-resolution Fourier Transform mass spectrometry (FIA-FTMS) on a QExactive Orbitrap (Thermo Fisher Scientific, Bremen, Germany), as described previously [19]. Multiplexed acquisition (MSX) was applied for the [M+NH4]+ of FC and the corresponding internal standard (FC[D7]). CE was recorded as [M+NH4]+ in positive ion mode in range m/z 500–1000 at a target resolution of 140,000 (at m/z 200). The accurate quantification of CE species by FIA-FTMS requires correction by individual response factors. The differences in response between CE species are because of structural features such as the length and double-bond number of acyl chains. The calculation of these CE species-specific response factors has been described in detail [19]. CE 17:0 and CE 22:0 (Sigma-Aldrich (Taufkirchen, Germany) were added as internal standards. 

### 2.3. Statistical Analysis

Data are shown as boxes, and the mean value ± standard deviation is given. Boxplots (minimum, maximum, median, first and third quartiles, small circles or asterisks above or below the boxes mark outliers) were used in the case that only one lipid/lipid ratio was shown.

The Mann–Whitney U-test, one-way ANOVA, Kruskal–Wallis-test or *t*-test were used (SPSS Statistics 25.0 program, IBM, Armonk, New York, NY, USA; and Microsoft Excel 2016, Redmond, Washington, DC, USA). A value of *p* < 0.05 after adjusting for multiple comparisons was regarded as significant. 

## 3. Results

### 3.1. Association of Age, Gender, Body Mass Index, Liver Steatosis and Diabetes with Serum FC and CE Species

One hundred seventy-eight patients with chronic HCV were included in the study, and fifteen different cholesteryl ester (CE) species as well as free cholesterol (FC) were measured in serum. Levels of these lipids did not differ between the 74 females and 104 males and did not correlate with age or body mass index (BMI) (see Appendix A). Accordingly, CE species and FC did not change with increasing body mass index (see Appendix A). The 74 patients with liver steatosis and the 104 patients without liver steatosis had similar concentrations of these CE species and FC (see Appendix A). Notably, levels of CE 20:5, 22:5 and 22:6 were reduced in the serum of the 20 diabetic patients (Figure 1A), and the relative concentrations (% of total CEs) of these CE species declined (*p* < 0.05). FC levels were not changed in HCV patients with diabetes (Appendix A), and thus, the CE/FC ratio was reduced (Figure 1B).

### 3.2. CE Species Levels and Distribution in Patients with Advanced Liver Disease 

There were 8 patients with diabetes in the group of HCV patients without cirrhosis, and 12 in the group of patients with cirrhosis. CE species did not differ between the non-diabetic and diabetic patients in these subgroups (data not shown). However, the number of patients was quite small and an effect of diabetes on serum CE levels cannot be ruled out. Thus, for the calculations of associations of CE species with measures of liver disease severity, HCV patients with diabetes were excluded. 

The fibrosis-4 (FIB4) score is an established measure to discriminate patients without and with liver cirrhosis [15]. The classification of patients with intermediate FIB-4 scores is, however, questionable [15]. CE 16:0, 18:2, 18:3, 20:3, 20:4 and 20:5 were reduced in patients with a high FIB-4 score in comparison to those with a low score. CE 18:2 and 20:4 were higher in the patients with intermediate FIB-4 scores than those with a high score (Figure 2A). 

In comparison to non-cirrhosis patients as determined by ultrasound examination, CE 16:0, 18:1, 18:2, 18:3, 20:3, 20:4, 20:5, 22:5 and 22:6 were reduced in the serum of patients with ultrasound-diagnosed liver cirrhosis (Figure 2B). Percent CE 14:0, 16:0 and 16:1 were higher and % CE 20:4, 20:5 and 22:6 were lower in the serum of cirrhosis patients, showing that the composition of CEs differs between these two groups (Figure 2C). In patients with a high FIB-4 score, % CE 16:0 and 16:1 were increased and % CE 18:2, 20:4 and 20:5 were reduced compared to patients with no fibrosis (Figure 2C). % CE 16:0 and 16:1 were higher and % CE 18:2 and 20:5 were lower in the fibrosis group compared to patients with intermediate scores (Figure 2C). Though the change of the CE profile was not identical when liver cirrhosis was diagnosed by ultrasound or the FIB-4 score, the increased proportion of CE 16:0 and 16:1 and the reduced proportion of CE 20:4 and 20:5 were observed in both groups (Figure 2C). 

FC was similar between patients with and without liver fibrosis/cirrhosis (*p* > 0.05). The CE/FC ratio was reduced in patients with a high FIB-4 score in comparison to the two groups with lower scores, and in patients with cirrhosis in comparison to non-cirrhosis patients (Figure 3A,B). 

### 3.3. Correlation of CE Species with the MELD Score and Laboratory Measures

In the patients without diabetes and without liver cirrhosis, there was a modest negative correlation of total CE levels with the MELD score (Figure 4A; r = −0.267, *p* = 0.032). However, none of the single CE species correlated with the MELD score (Table 1). CE 18:2 was negatively linked with international normalized ratio (INR). Albumin, ALT, AST, creatinine, bilirubin, CRP and leukocytes did not correlate with any of the CE species (Table 1).

In the group of patients with liver cirrhosis (28 patients because the 12 patients with diabetes were excluded), CE 18:2, 20:3, 20:4 and 22:6 as well as total CE levels (Figure 4B; r = −0.564, *p* = 0.030) negatively correlated with the MELD score. CE 18:2, 20:3, 20:4 and 22:6 negatively correlated with the INR, and CE 18:2, 20:4 and 22:6 with bilirubin. Albumin was positively linked with CE 16:0, 18:2, 20:4 and 22:6. ALT, AST, CRP, leukocytes and creatinine did not correlate with any CE species (Table 2). 

### 3.4. Effect of DAA Therapy for HCV on Serum CE Species 

The rise in total CE levels at 4 and 12 weeks after the start of DAA therapy was noticed in patients without and with liver cirrhosis (Figure 5A,B). FC did not increase during therapy in the two subgroups (Figure 5A,B). The CE/FC ratio was higher at 4 and 12 weeks after therapy start in both groups (Figure 5C,D). Notably, the CE/FC ratio at 4 and 12 weeks after therapy start was still higher in patients without cirrhosis than those with liver cirrhosis (Figure 5C,D).

All but CE 14:0, 14:1, 15:1 and 16:1 were increased at 4 weeks after therapy start in comparison to pre-treatment levels in the non-cirrhosis group (Figure 6A). At therapy end, CE 15:0, 16:0, 18:1, 18:2, 18:3, 20:3, 20:4, 20:5 and 22:6 were higher than the levels in the serum of the patients before treatment (Figure 6A). The ratio of CE 18:0/16:0 is a marker of elongation of long-chain fatty acids family member 6 (ELOVL6) activity. The ratio CE 18 to CE 16 was modestly higher at therapy end (*p* = 0.02). % CE 16:0 was lower at 4 and 12 weeks after therapy start and % CE 18:3 was higher at these time points when compared to pre-treatment levels. The rise in % CE20:3 occurred at week 4 after therapy start and returned to pre-treatment levels at therapy end (Figure 6B). None of the CE species were significantly increased at 4 or 12 weeks after therapy start in the cirrhosis group (Figure 6C). 

### 3.5. Association of CE Species with Non-Invasive Measures of Liver Cirrhosis at Therapy End 

At therapy end, patients with diabetes in comparison to non-diabetic patients had lower levels of CE 20:4 (*p* = 0.008), and the relative content of CE 20:4 was also reduced (*p* = 0.011). Thus, patients with diabetes were excluded from the calculations. According to the FIB-4 score, 20 patients had cirrhosis, 31 patients had intermediate levels and 103 patients did not have liver fibrosis at therapy end. CE 16:0, 18:1, 18:2, 18:3, 20:3, 20:4, 20:5 and 22:6 were reduced in the serum of patients with a high FIB-4 score in comparison to patients with a low score (Table 3). 

In the 29 patients with ultrasound-diagnosed liver cirrhosis, serum CE 16:0, 18:1, 18:2, 18:3, 20:3, 20:4, 20:5 and 22:6 were lower than in the patients without cirrhosis (Table 4). 

Notably, the proportion of CE 14:0, 16:0 and 16:1 was higher and that of CE 20:3, 20:4 and 22:6 was lower in patients with a high FIB4 score compared to patients with a low score (Figure 7A). This suggests that fatty acid elongation may be also impaired. The ratio of CE 18:0/16:0 was lower in patients with a high FIB-4 score compared to those with a low score (Figure 7B). In ultrasound-diagnosed cirrhosis, % CE 14:0, 16:0 and 16:1 were induced and % CE 20:4, 20:5 and 22:6 declined in comparison to patients without cirrhosis (Figure 7C), and the CE 18:0/16:0 ratio was reduced (Figure 7D).

Negative correlations with the MELD score in patients with ultrasound-diagnosed liver cirrhosis were found with CE 16:0 (r = −0.627, *p* = 0.004), 18:1 (r = −0.603, *p* = 0.008), 18:2 (r = −0.703, *p* < 0.001), 18:3 (r = −0.590, *p* = 0.011), 20:3 (r = −0.741, *p* < 0.001), 20:4 (r = −0.827, *p* < 0.001) and 22:6 (r = −0.691, *p* = 0.001). In patients without liver cirrhosis, none of the CE species was associated with the MELD score (*p* > 0.05).

### 3.6. Association of CE Species with Viral Titer and Genotype

CE species and FC did not correlate with viral load in the whole cohort, and in the HCV patients without and with ultrasound-diagnosed cirrhosis (Appendix A and data not shown).

Associations with genotype were calculated after the exclusion of diabetic patients and patients with liver cirrhosis. Forty-one patients had genotype 1a, forty-eight 1b, twenty-eight 3a and thirteen had a different genotype and were grouped together (rare). Here, genotype 3a infection led to lower CE 15:0 and 16:0 compared to 1a, and lower CE 16:0, 18:2 and 20:4 compared to 1b (Figure 8). The relative content of CEs did not differ between the genotypes with the exception of CE15:0, which was lower in genotype 3a compared to 1a (*p* = 0.02). These genotype-associated variations disappeared at therapy end (data not shown). 

### 3.7. Correlation of CE Species with HDL and LDL

In the group of HCV patients without liver cirrhosis, all but CE 22:4 positively correlated with LDL before therapy. CE 18:2 and 22:4 were positively associated with HDL levels. In cirrhosis, CE 15:0, 16:0, 18:1, 18:2, 18:3, 20:3, 20:4, 20:5 and 22:6 correlated with LDL and CE 15:0, 18:2, 22:4 and 22:5 with HDL (Table 5).

At therapy end, all CE species positively correlated with LDL in the non-cirrhosis group. Associations with HDL were not significant. In cirrhosis, CE 16:0, 18:1, 18:2, 18:3, 20:3, 20:4, 20:5 and 22:6 correlated with LDL and CE 14:0 and 22:5 with HDL (Table 5).

## 4. Discussion

This study shows that DAA therapy increases serum CE levels and leads to a more favorable CE profile. Patients with liver cirrhosis display an adverse CE profile, the relative content of saturated and monounsaturated CEs with short acyl chains is high and that of polyunsaturated CE species with 20 or 22 carbon atoms is low, and this does not change by DAA therapy.

Nowadays, it is well accepted that the effective elimination of HCV causes a rise in serum LDL and cholesterol [17,47,48,49,50]. All but CE 14:0, CE 14:1, CE 15:1 and CE 16:1 were significantly increased at therapy end.

LCAT forms CE 20:4, 22:5 and 22:6, whereas CE 16:0, 18:1 and 18:3 are derived from liver ACAT [23,24,25] (Figure 9). This might indicate that ACAT as well as LCAT activity are higher at therapy end—a hypothesis that has to be yet confirmed. However, various other enzymes and pathways have a role in CE composition. Cholesterol-esterifying activity in serum was found to be decreased in patients with acute hepatitis and chronic alcoholic liver disease, and besides LCAT, CE hydrolase seems to have a function herein [51]. Thus, it is currently unclear which pathways are involved. Notably, % CE 16:0 was reduced and % CE 18:3 was induced at therapy end, suggesting that a more favorable CE profile exists when HCV is efficiently cleared.

HCV patients have a higher incidence of metabolic diseases, including atherosclerosis [2]. There is evidence that cardiovascular diseases and insulin resistance improve after SVR by DAA therapy [52]. Future studies have to evaluate whether the decline of % CE 16:0 and the higher abundance of % CE 18:3 contribute to these beneficial effects.

The recovery of the CE species in the cirrhosis group was not significant. Total CE levels similarly increased during therapy in cirrhosis and non-cirrhosis patients, and were 119% and 116% higher at therapy end, respectively. This suggests that the elimination of HCV changes the CE levels of both groups. Because of the lower number of patients with cirrhosis, this was not significant in this subgroup.

Viral load did not correlate with FC or any of the CE species. Interestingly, genotype 3 infection reduced CE 15:0 and 16:0 in comparison to 1a infection, and CE 16:0, 18:2 and 20:4 in comparison to 1b infection. These differences did not persist at therapy end, showing that genotype 3 differently affects CE composition in comparison to genotype 1. Genotype 3-infected patients were described to have lower serum LDL cholesterol than genotype 1-infected patients [53]. Yet, CEs such as cholesteryl linoleate were higher in genotype 3 than genotype 1 [53]. The present analysis revealed lower levels of CE 18:2 in genotype 3- than 1b-infected patients, and currently, there is no explanation for these divergent results. The two studies agree that the genotype-related changes of the CE lipidome are not apparent in patients who achieved SVR [53].

Besides having low levels of serum cholesterol, patients with liver cirrhosis have a reduced CE/FC ratio. This may be due to a lower activity of enzymes involved in FC esterification. Cholesterol esterification in plasma has been described as a marker for liver function in patients with advanced stages of liver disease [35]. Interestingly, the CE/FC ratio increased during DAA treatment in the cirrhosis and non-cirrhosis group, showing that the cholesterol esterification rate is higher in both cohorts. Liver function does not greatly improve during DAA therapy [7,8] and the MELD score of our patient group did not change [46], indicating that the cholesterol esterification rate is not solely affected by liver disease severity.

Notably, serum FC levels did not differ between cirrhosis and non-cirrhosis patients before and after therapy. Pathways contributing to serum FC levels such as reverse cholesterol export [54] are, thus, not grossly impaired in cirrhosis.

In particular, CEs with longer acyl chains declined in the serum of patients with liver cirrhosis. In patients with a high FIB-4 score, CE 16:0, 18:2, 18:3, 20:3, 20:4 and 20:5 were low (Figure 9). These species, and CE 18:1, 20:5, 22:5 and 22:6, were reduced in the serum of patients with ultrasound-diagnosed liver cirrhosis. Moreover, CE 18:2, 20:3, 20:4 and 22:6 negatively correlated with the MELD score in patients with liver cirrhosis, further indicating an association with liver disease severity.

Shorter CEs are the product of ACAT and the longer CEs from LCAT. Because CE 18:1 as well as CE 22:5 were low in cirrhosis, both cholesterol esterification pathways may be impaired (Figure 9). Decreased LCAT activity has been described in patients with liver cirrhosis [29], but whether ACAT2 activity is also low has not been clarified yet. In addition, ELOVL6, which catalyzes the elongation of saturated and monounsaturated fatty acids with 12 to 16 carbon atoms, was found to be lower expressed in patients with advanced fibrosis [55] (Figure 9). Impaired activity of this enzyme may partly explain the depletion of longer chain fatty acids in CEs, and the CE 18-/C 16 ratio is low in our HCV patients with liver cirrhosis. Notably, the CE 18/CE 16 ratio was modestly higher at the end of DAA therapy, suggesting that HCV infection may also lower the activity of ELOVL6 (Figure 9). The different CE species are, however, not reduced to a similar extent in the serum of patients with liver cirrhosis. The proportion of CEs with short acyl chains and no or one double bond increased, whereas CEs with longer acyl chains and at least two double bonds declined. This did not change at therapy end, showing that liver cirrhosis is associated with a unfavorable CE profile. In patients with decompensated liver cirrhosis mostly because of alcohol abuse, the relative content of total plasma CE 14:0, 16:0 and 18:1 was higher and the relative content of CE 18:2 and 20:4 was lower compared to controls [56], and thus, were similarly altered, as observed in the HCV cohort studied herein. The activity of delta-6-desaturase, an enzyme involved in the synthesis of polyunsaturated fatty acids, was found to be reduced in liver cirrhosis [57], and this may also partly explain this observation.

Liver cirrhosis is not associated with a higher risk for atherosclerosis [58] and the pathophysiological role of the altered CE profile has still to be evaluated. A frequent comorbidity of liver cirrhosis is type 2 diabetes [59], but the associations of serum CE species with glucose homeostasis have not been finally resolved.

HCV-infected patients with diabetes studied herein had lower serum levels of CE 20:5, 22:5 and 22:6, and % CE 20:5 and 22:6 were reduced. In patients with type 2 diabetes, the proportions of CE 18:0 and CE 20:3n−6 were higher, and those of CE 18:1n−7 and C20:4n−6 were reduced compared to patients with normal glucose metabolism [60]. A further study identified protective associations of CE 18:1n−7 and 18:1n−9 and harmful associations of CE 18:3n−6 and 18:0 with insulin sensitivity and beta-cell function [61]. Current findings on the association of CE species with diabetes are inconsistent and further research is needed. It has to be noted that—in the current analysis—gender, BMI and age were not associated with FC levels or the change in any of the CE species analyzed.

In serum, CE species are carried by LDL, HDL and VLDL. LDL cholesterol content is about 2-fold higher than that of VLDL and HDL [20]. CE levels in serum mainly correlated with LDL levels, and thus, serum CE content was mostly related to LDL rather than HDL levels.

Interestingly, CE composition did not greatly differ between HDL, LDL and VLDL of healthy volunteers [20] and patients with liver cirrhosis [56]. This suggests that the CE profile of LDL, HDL and VLDL is changed in cirrhosis. How and whether an altered CE c influences the function of lipoproteins needs future investigations.

This study has limitations. Total serum CE and FC levels were measured but the composition of individual lipoproteins and LCAT activity were not analyzed. Serum was not collected in the fasted state, and dietary habits of the patients were not documented. However, serum cholesterol levels do not greatly vary during the day [62]. It is, moreover, very unlikely that the identified changes in CE composition achieved by HCV therapy and in patients with liver cirrhosis are explained by different diets. A further limitation is that healthy controls and patients with liver diseases of distinct etiologies were not included.

In summary, the present analysis showed that viral genotype, DAA therapy and cirrhosis differentially affect serum CE species levels. Prospective studies have to evaluate the prognostic value of CE species for cardiovascular diseases, insulin resistance, morbidity and mortality in chronic HCV infection and liver cirrhosis.

## Figures and Tables

**Figure 1 biomedicines-10-03152-f001:**
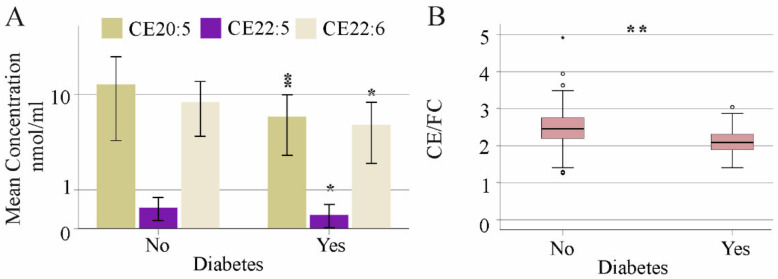
Serum cholesteryl ester (CE) species in relation to diabetes in patients with chronic HCV. (**A**) CE species, which differ in patients with and without diabetes. (**B**) CE/FC ratio of patients with and without diabetes * *p* < 0.05, ** *p* < 0.01. Small circles above the boxes mark outliers.

**Figure 2 biomedicines-10-03152-f002:**
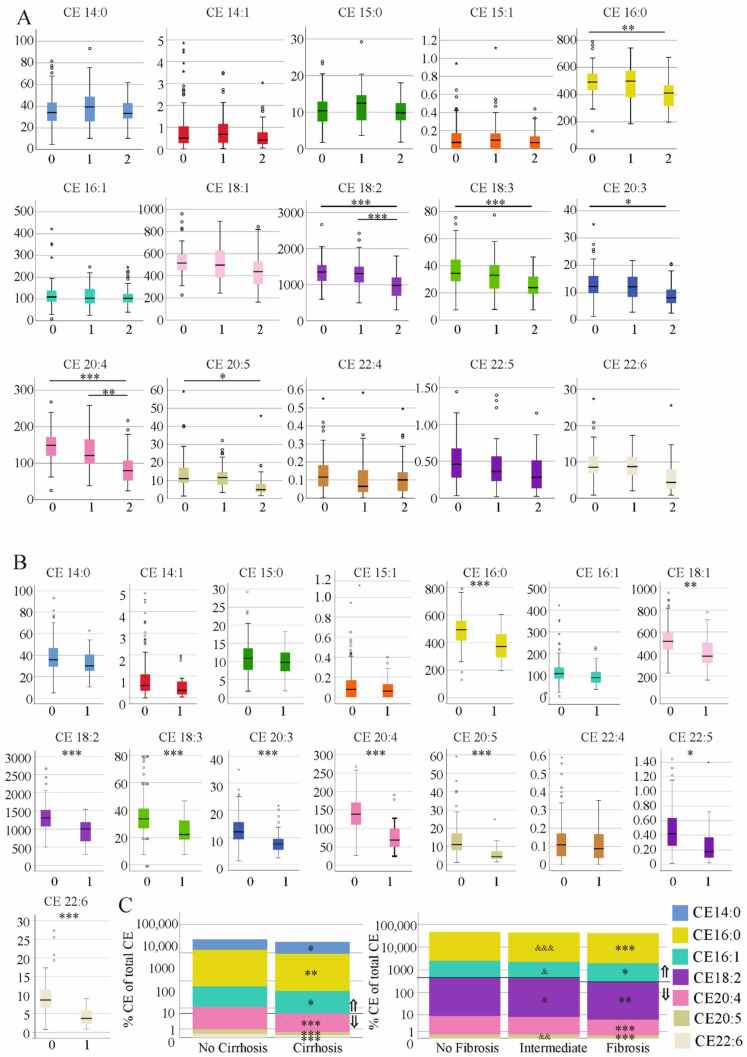
Cholesteryl ester (CE) species and CE profile in relation to the FIB-4 score and to cirrhosis diagnosed by ultrasound. (**A**) CE species in patients with no fibrosis (0), in patients with an intermediate FIB-4 score (1) and patients with fibrosis (2). (**B**) CE species in patients without and with liver cirrhosis diagnosed by ultrasound. Small circles or asterisks above or below the boxes mark outlier. (**C**) % CE of total CE levels of the patients described in (**A**,**B**) ⇑ higher in fibrosis/cirrhosis ⇓ lower in fibrosis/cirrhosis (& *p* < 0.05, && *p* < 0.01; &&& *p* < 0.001 for comparison of patients with an intermediate and a high FIB-4 score. * *p* < 0.05, ** *p* < 0.01, *** *p* < 0.001.

**Figure 3 biomedicines-10-03152-f003:**
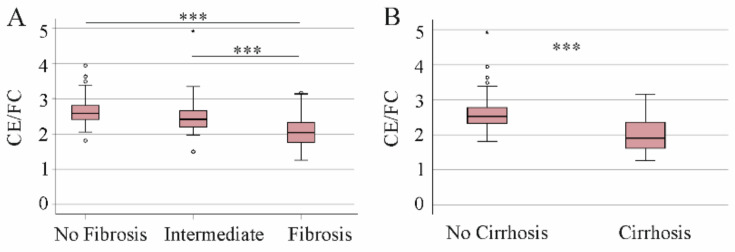
Cholesteryl ester (CE)/free cholesterol (FC) ratio in relation to the FIB-4 score and to cirrhosis diagnosed by ultrasound. (**A**) CE/FC ratio in the patients with no fibrosis, intermediate values and patients with fibrosis defined by the FIB-4 score. (**B**) CE/FC ratio in the patients without and with liver cirrhosis diagnosed by ultrasound. Small circles or asterisks above or below the boxes mark outlier. *** *p* < 0.001.

**Figure 4 biomedicines-10-03152-f004:**
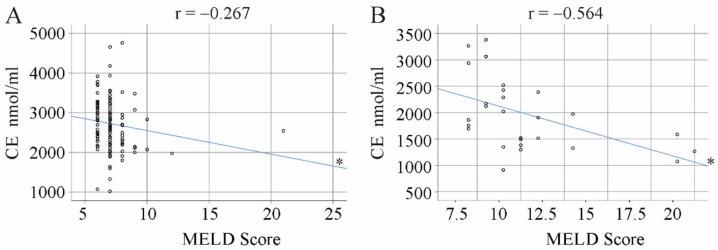
Correlation of total cholesteryl ester (CE) levels with the MELD score. (**A**) Patients without liver cirrhosis, as diagnosed by ultrasound. (**B**) Patients with ultrasound-diagnosed liver cirrhosis. The blue line is the correlation line. * *p* < 0.05.

**Figure 5 biomedicines-10-03152-f005:**
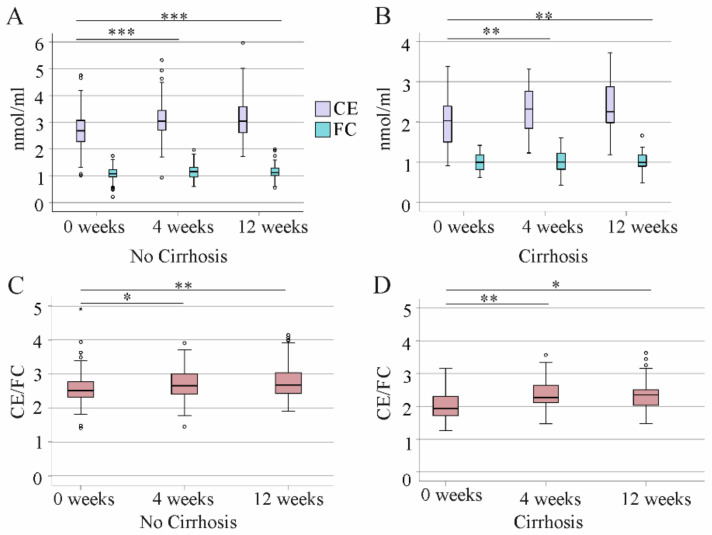
Cholesteryl ester (CE) and free cholesterol (FC) levels during the study. (**A**) Concentrations in serum of patients without cirrhosis diagnosed by ultrasound. (**B**) Concentrations in serum of patients with cirrhosis diagnosed by ultrasound. (**C**) CE/FC ratio in serum of patients without cirrhosis diagnosed by ultrasound. (**D**) CE/FC ratio in serum of patients with cirrhosis diagnosed by ultrasound. * *p* < 0.05, ** *p* < 0.01, *** *p* < 0.001. Small circles or asterisks above or below the boxes mark outlier.

**Figure 6 biomedicines-10-03152-f006:**
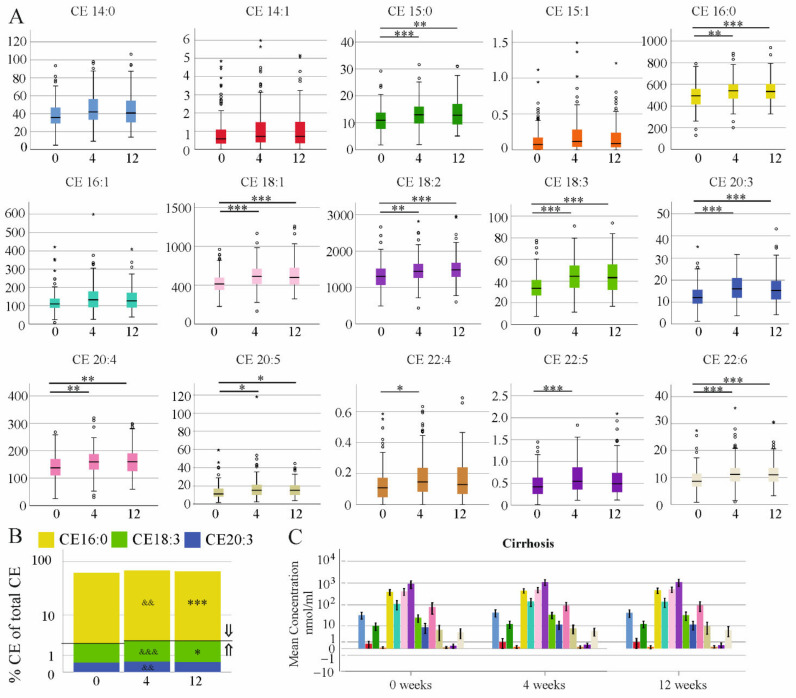
Cholesteryl ester (CE) species during the study. (**A**) Concentrations in serum of patients without cirrhosis diagnosed by ultrasound before therapy, and at 4 and 12 weeks after therapy start. Small circles or asterisks above or below the boxes mark the outlier. * *p* < 0.05, ** *p* < 0.01, *** *p* < 0.001. (**B**) % CE of total CEs in the serum of patients without cirrhosis diagnosed by ultrasound before therapy, and at 4 and 12 weeks after therapy start. ⇑ higher compared to 0 weeks, ⇓ lower compared to 0 weeks (&& *p* < 0.01, &&& *p* < 0.001 for comparison of 0 and 4 weeks and * *p* < 0.05, *** *p* < 0.001 for comparison of 0 and 12 weeks). (**C**) Concentrations in the serum of patients with cirrhosis diagnosed by ultrasound.

**Figure 7 biomedicines-10-03152-f007:**
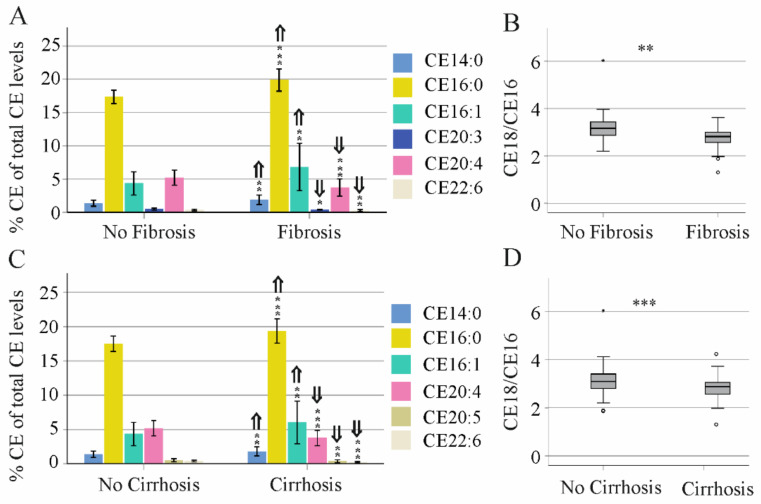
Proportion of cholesteryl ester (CE) species in relation to the FIB-4 score and to cirrhosis diagnosed by ultrasound. (**A**) % CE of total CE in patients with no fibrosis and patients with fibrosis defined by the FIB-4 score. (**B**) CE 18/CE 16 ratio in serum of these patients. (**C**) % CE of total CE in patients without and with liver cirrhosis diagnosed by ultrasound. (**D**) CE 18/CE 16 ratio in serum of these patients. * *p* < 0.05, ** *p* < 0.01, *** *p* < 0.001. ⇑ higher compared to no fibrosis/no cirrhosis, ⇓ lower compared to no fibrosis/no cirrhosis.

**Figure 8 biomedicines-10-03152-f008:**
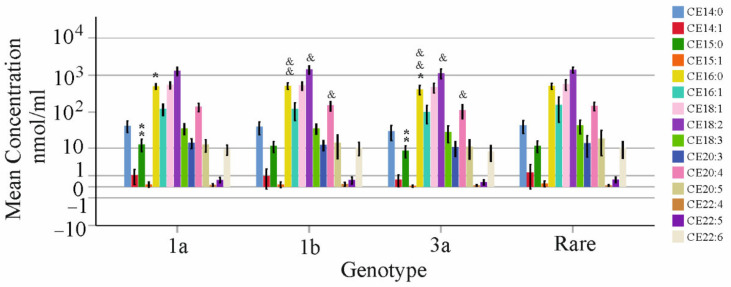
Cholesteryl ester (CE) species in relation to viral genotype before therapy. CE species in the serum of patients without liver cirrhosis before therapy. * *p* < 0.05, ** *p* < 0.01 for comparison of genotype 1a and 3a, and & *p* < 0.05, && *p* < 0.01 for comparison of genotype 1b and 3a.

**Figure 9 biomedicines-10-03152-f009:**
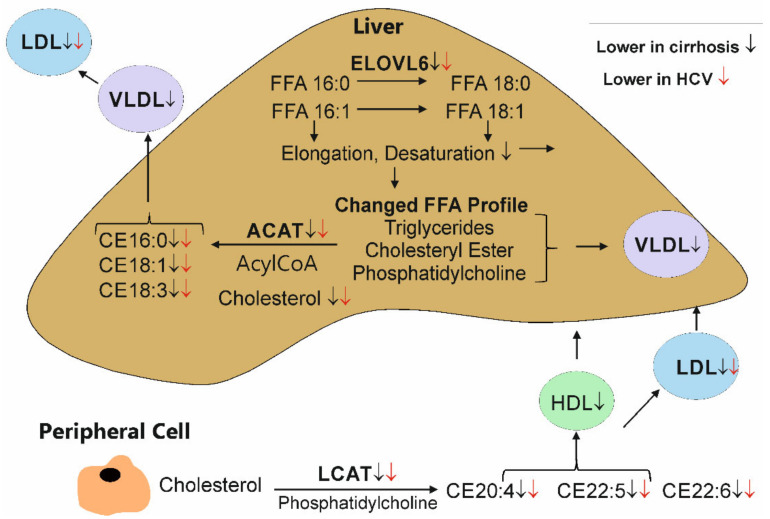
Pathways which may contribute to the altered CE profile in HCV and liver cirrhosis. Free cholesterol in the liver is converted to CEs by ACAT, which produces atherogenic CE species. These are released from the liver by VLDL, which is converted to LDL in the circulation. Elongation of very-long chain fatty acids (ELOVL) 6 was found to belower expressed in the fibrotic liver, and activity of further elongases and desaturases may be impaired. Free fatty acids (FFA) are used for the synthesis of various lipid classes such as triglycerides, diacylglycerols, cholesteryl ester and phosphatidylcholine. LCAT in serum uses phosphatidylcholine to esterify cholesterol derived from peripheral cells and tissues. Here, highly unsaturated CE species are being formed. The CE species (found in the current work to be low when cirrhosis was defined by ultrasound), lipoproteins and enzymes which are reduced in cirrhosis are marked with a black arrow. A red arrow marks CE species and lipoproteins, which are low in HCV infection. Current findings are in agreement with improved ELOVL6, ACAT and LCAT activity at the end of DAA therapy. This is in line with a higher CE/FC ratio at therapy end. Whether the levels of these enzymes are reduced or FFA availability is limited must be experimentally clarified. ↓ Lower in cirrhosis, ↓ Lower in HCV.

**Table 1 biomedicines-10-03152-t001:** Spearman correlation coefficients of CE species and the MELD score as well as laboratory measures of patients without diabetes and without liver cirrhosis before therapy. * *p* < 0.05.

CE	MELD	INR	Albumin	ALT	AST	Creatinine	Bilirubin	CRP	Leukocytes
14:0	−0.082	−0.066	−0.113	−0.037	0.024	0.040	0.008	−0.061	−0.115
14:1	−0.080	−0.076	−0.134	−0.013	0.022	0.021	−0.058	0.030	−0.014
15:0	−0.108	−0.114	−0.008	−0.124	−0.110	−0.030	0.034	0.024	−0.180
15:1	−0.093	−0.048	−0.197	−0.166	−0.134	−0.006	0.089	0.068	−0.114
16:0	−0.212	−0.254	0.048	−0.095	−0.087	0.116	0.083	0.076	−0.079
16:1	−0.123	−0.088	−0.099	−0.070	0.056	−0.042	0.003	0.166	0.088
18:1	−0.198	−0.220	0.011	−0.159	−0.119	0.036	0.054	0.087	0.101
18:2	−0.231	−0.282 *	0.149	−0.121	−0.172	0.027	0.105	0.105	−0.176
18:3	−0.243	−0.243	−0.117	−0.193	−0.200	0.098	−0.026	0.148	−0.021
20:3	−0.121	−0.154	−0.051	−0.006	−0.006	0.031	0.017	−0.012	0.025
20:4	0.176	−0.255	−0.015	−0.102	−0.164	0.117	0.106	0.175	−0.049
20:5	−0.222	−0.214	−0.063	−0.146	−0.186	0.073	0.071	0.261	0.008
22:4	−0.061	−0.108	−0.021	−0.071	−0.144	0.108	0.161	0.063	−0.147
22:5	0.025	0.022	−0.198	0.019	−0.026	0.004	0.161	0.026	0.010
22:6	−0.120	−0.149	0.054	−0.004	−0.029	0.062	0.116	0.200	−0.133

**Table 2 biomedicines-10-03152-t002:** Spearman correlation coefficients of CE species with the MELD score as well as laboratory measures of patients without diabetes and with liver cirrhosis before therapy. * *p* < 0.05, ** *p* < 0.01, *** *p* < 0.001.

CE	MELD	INR	Albumin	ALT	AST	Creatinine	Bilirubin	CRP	Leukocytes
14:0	−0.225	−0.332	0.276	0.100	0.005	−0.347	−0.128	−0.023	0.221
14:1	−0.270	−0.350	0.209	0.179	0.011	−0.201	−0.112	0.041	0.425
15:0	−0.261	−0.362	0.377	0.176	−0.044	−0.182	−0.125	0.146	0.349
15:1	−0.175	−0.231	0.075	0.199	0.067	−0.161	−0.015	0.042	0.319
16:0	−0.494	−0.530	0.639 **	0.010	−0.268	−0.313	0.424	−0.130	0.389
16:1	−0.172	−0.235	0.048	−0.011	0.066	−0.065	0.017	0.095	0.483
18:1	−0.484	−0.524	0.510	0.084	−0.139	−0.235	−0.377	−0.158	0.400
18:2	−0.593 *	−0.573 *	0.650 **	0.105	−0.252	−0.315	−0.578 *	−0.189	0.419
18:3	0.434	−0.507	0.337	0.108	−0.046	−0.223	−0.251	−0.117	0.468
20:3	−0.616 **	−0.604 *	0.482	0.253	−0.030	−0.237	−0.476	−0.073	0.521
20:4	−0.739 ***	−0.733 ***	0.744 ***	0.130	−0.226	−0.258	−0.610 **	−0.162	0.524
20:5	−0.468	0.519	0.519	0.031	−0.271	−0.262	−0.375	0.029	0.372
22:4	−0.310	−0.289	0.237	0.061	0.098	−0.058	−0.132	0.060	0.391
22:5	−0.327	−0.331	0.388	0.102	−0.043	−0.218	−0.123	0.085	0.312
22:6	−0.702 ***	−0.708 ***	0.693 **	0.197	−0.150	−0.270	−0.574 **	−0.118	0.472

**Table 3 biomedicines-10-03152-t003:** CE levels in serum of patients with a low and a high FIB-4 score at therapy end. Only species which significantly differed between the two groups are listed. ** *p* < 0.01, *** *p* < 0.001.

		No Fibrosis		Fibrosis			
CE nmol/mL	Median	Minimum	Maximum	Median	Minimum	Maximum	*p*-Value
16:0	538.29	349.67	873.71	398.63	280.04	697.76	**
18:1	591.08	381.86	1258.74	469.63	272.86	760.45	***
18:2	1496.90	845.63	2926.86	856.50	312.47	1911.60	***
18:3	45.55	16.66	93.63	28.69	11.51	49.09	***
20:3	15.27	4.24	43.19	8.65	2.64	15.20	***
20:4	159.33	77.53	296.05	77.58	27.60	180.98	***
20:5	14.57	3.65	44.58	6.20	1.41	39.61	**
22:6	10.96	3.77	30.68	4.59	0.92	18.33	***

**Table 4 biomedicines-10-03152-t004:** CE levels in serum of patients without and with liver cirrhosis diagnosed by ultrasound at therapy end. Only species which significantly differed between the two groups are listed. * *p* < 0.05, ** *p* < 0.01, *** *p* < 0.001.

	No Cirrhosis			Cirrhosis			
CE nmol/mL	Median	Minimum	Maximum	Median	Minimum	Maximum	*p*-Value
16:0	536.44	328.26	939.53	427.61	280.04	697.76	*
18:1	600.85	323.54	1258.74	494.44	272.86	750.15	**
18:2	1480.27	603.00	2955.23	1045.26	312.47	1911.60	***
18:3	44.10	16.66	93.63	30.26	11.51	69.24	**
20:3	15.15	4.24	43.19	10.32	2.64	21.97	**
20:4	159.91	60.10	298.83	95.27	27.60	180.98	***
20:5	14.99	3.65	44.58	7.26	1.41	19.71	***
22:6	11.03	3.69	30.68	5.18	0.92	13.16	***

**Table 5 biomedicines-10-03152-t005:** Spearman correlation coefficients for the association of CE species with LDL and HDL before therapy and at therapy end in patients with and without ultrasound-diagnosed liver cirrhosis. * *p* < 0.05, ** *p* < 0.01, *** *p* < 0.001.

	Before Therapy				Therapy End			
	No Cirrhosis		Cirrhosis		No Cirrhosis		Cirrhosis	
CE	LDL	HDL	LDL	HDL	LDL	HDL	LDL	HDL
14:0	0.514 ***	0.149	0.439	0.554	0.559 ***	0.157	0.440	0.651 **
14:1	0.449 ***	0.013	0.393	0.336	0.306 ***	0.075	0.429	0.501
15:0	0.523 ***	0.263	0.642 **	0.626 **	0.537 ***	0.247	0.544	0.539
15:1	0.408 ***	0.234	0.310	0.345	0.360 **	0.213	0.427	0.389
16:0	0.752 ***	0.246	0.862 ***	0.554	0.878 **	0.119	0.799 ***	0.414
16:1	0.464 ***	0.157	0.415	0.205	0.447 ***	0.193	0.438	0.539
18:1	0.585 ***	0.184	0.821 ***	0.510	0.707 ***	0.169	0.773 ***	0.433
18:2	0.745 ***	0.348 **	0.884 ***	0.563 *	0.791 ***	0.229	0.838 ***	0.38
18:3	0.597 ***	0.220	0.684 **	0.328	0.562 ***	0.149	0.639 **	0.465
20:3	0.593 ***	0.047	0.723 ***	0.485	0.679 ***	0.038	0.727 ***	0.483
20:4	0.616 ***	0.208	0.820 ***	0.506	0.576 ***	−0.016	0.701 **	0.401
20:5	0.498 ***	0.250	0.684 **	0.266	0.388 ***	0.160	0.589 *	0.244
22:4	0.244	0.390 ***	0.415	0.574 *	0.292 **	−0.043	0.290	0.086
22:5	0.306 **	0.159	0.237	0.631 **	0.288 **	0.141	0.248	0.553 *
22:6	0.543 ***	0.147	0.825 ***	0.464	0.473 ***	0.088	0.712 ***	0.247

## Data Availability

The datasets generated and/or analyzed during the current study are available from the corresponding author on request.

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
