# Peer review of "HCV Infection and Liver Cirrhosis Are Associated with a Less-Favorable Serum Cholesteryl Ester Profile Which Improves through the Successful Treatment of HCV"

_biomedicines, 2022, doi:10.3390/biomedicines10123152_

Round 1

Reviewer 1 Report

Authors aimed to find out whether the CE profile differs between HCV patients with and without liver cirrhosis, and to study the effect of DAA therapy on CE composition of both groups. This is an interesting topic. There are several major concerns.

Major

1) What it the theoretical rationale of change in CE characteristics in patients with cirrhosis?

2) In the clinical practice, after successful DAA therapy, many patients remained having fatty liver. Is there any explanation for this finding?

Minor

1) Several related articiels, e.g. PMID: 34255961, PMID: 33601868, PMID: 33317245, should be cited in the introduction part.

Author Response

We thank the reviewer for the highly valuable comments, which helped us to improve our manuscript.

We changed the title and replaced “restored” by “improved” because we do not have a healthy control group for comparison. 

Major

  • What it the theoretical rationale of change in CE characteristics in patients with cirrhosis?

As is already discussed ACAT and LCAT may have a role. We now added data suggesting that ELVOVL6 activity is reduced. Please see figure 7b and c. We also added figure 9 as a summary. This is, however, an observational study not well suited to define the rationale of any changes.

2) In the clinical practice, after successful DAA therapy, many patients remained having fatty liver. Is there any explanation for this finding?

This is now shortly explained at the beginning of the Introduction. The metabolic syndrome and fatty liver disease are very common in the “normal” population, and about 30% are affected. Thus, compared to viral steatosis non-viral caused steatosis will not resolve after successful therapy.   

Minor

  • Several related articiels, e.g. PMID: 34255961 (now cited, reference 10), PMID: 33601868 (cited, reference 5) PMID: 33317245 (cited, line 45), should be cited in the introduction part.

These articles have been cited. PMID: 34255961 (now cited, lane 48 to 52), PMID: 33601868 (cited, line 45), and PMID: 33317245 (cited, Reference 6)

Reviewer 2 Report

The manuscript submitted by Weigand et al is an interesting study that has investigated the circulating  concentrations of cholesterol esters (CE) in patients with hepatitis C virus (HCV) and associated liver fibrosis. The text is quite dense and there is no real narrative or rationale for the work. As it stands, this is an observational study. There are a number of points, which the authors should attempt to address:

Major Corrections

1. It would be useful to see a figure showing the different pathways of CE metabolism and an indication if there are pathway specific changes in the stratified cohorts.

2. Is the intention to use the CE profiles as a marker of disease progression or is the aim to understand the biology of HCV progression and effect of treatment?

3. On several occasions the authors use the phrase ‘favourable’ or ‘adverse’ CE profile, but this is not defined and it is not clear what the parameters for a ‘good’ or ‘bad’ profile is. In general, the figures, particularly the graphs should be improved as they are difficult to interpret.

Minor Corrections

1. Abstract: what does a ‘less favourable CE profile’ mean? This is quite a subjective statement.

2. Introduction: The fibrosis-4 (FIB-4) 55 score is calculated from age, aspartate aminotransferase (AST), alanine aminotransferase 56 (ALT) and platelet count [8]. Is it concentration or activity of the enzymes that has been measured?

3. Line 70: Hepatocytes 70 express ACAT2, and ACAT2 – should be ‘expressing’.

4. Line 76-79 – include with preceding paragraph.

5. Line 100 should begin ‘The aim…’.

6. Materials and Methods: Define the size of the study cohort. How many were recruited and what was the drop-out rate?

7. Line 131: What does this mean, ‘corrected for their species-specific response’?

8. Results: It would have been helpful to see data from a non-HCV control group.

9. What is the purpose of Figure 1B?

10. What is the relevance of diabetes to HCV? This is not clear. It should be introduced before the results section.

11. Figure 2 could be presented better. It is difficult to interpret along with Figure 6.

12. Discussion: Again, a favourable and adverse CE profile is mentioned but at no point is this defined. What makes a CE profile ‘good’ or ’bad’?

Author Response

We thank the reviewer for the highly valuable comments which helped us to improve our manuscript.

We changed the title and replaced “restored” by “improved” because we do not have a healthy control group for comparison. 

Major Corrections

  1. It would be useful to see a figure showing the different pathways of CE metabolism and an indication if there are pathway specific changes in the stratified cohorts.

This is an observational study not appropriate to identify specific pathways. To address this question we have added figure 9. We also calculated the CE 18 / CE 16 ratio which may be related to Elovl9 activity and was shown to decline in patients with liver cirrhosis (Figure 7B, D).

  1. Is the intention to use the CE profiles as a marker of disease progression or is the aim to understand the biology of HCV progression and effect of treatment?

Main aim was the study the effect of DAA therapy on the CE profile. HCV elimination is associated with higher LDL but improved insulin resistance and lower risk for cardiovascular diseases. This is now better explained at the end of the Introduction.

  1. On several occasions the authors use the phrase ‘favourable’ or ‘adverse’ CE profile, but this is not defined and it is not clear what the parameters for a ‘good’ or ‘bad’ profile is. In general, the figures, particularly the graphs should be improved as they are difficult to interpret.

This is now better explained at the end of the Introduction.

“Further studies showed that ACAT2 derived CE species are predominantly atherogenic blood lipids [10]. LCAT overexpression resulted in higher HDL levels and prevented the development of diet-induced atherosclerosis [19]. Higher unsaturation of LDL carried CEs was supposed to have beneficial cardiovascular effects [22].” Paragraph already included in the previous manuscript.

“Effective therapy of HCV elevates serum LDL and cholesterol levels, which are risk factors for cardiovascular diseases [39, 40]. Elimination of HCV is, however, associated with a lower risk for cardiovascular diseases [3]. Not all of the CE species seem to contribute to an increased risk, and a higher polyunsaturated to saturated CE ratio protected from atherosclerosis [39]. Among the lipids with a strong predictive value for cardiovascular diseases were CE species with a low carbon number and a low double-bond content [41]. “ newly added text.

Minor Corrections

  1. Abstract: what does a ‘less favourable CE profile’ mean? This is quite a subjective statement.

This is now explained in the text.

  1. Introduction: The fibrosis-4 (FIB-4) 55 score is calculated from age, aspartate aminotransferase (AST), alanine aminotransferase 56 (ALT) and platelet count [8]. Is it concentration or activity of the enzymes that has been measured?

The activity of these enzymes has been measured (U/l) and is also used for calculation. This is now given in the Introduction (please see line 74 - 75).

  1. Line 70: Hepatocytes 70 express ACAT2, and ACAT2 – should be ‘expressing’.

We have rewritten the sentence for clarity.

  1. Line 76-79 – include with preceding paragraph.

This was corrected.

  1. Line 100 should begin ‘The aim…’.

This was corrected.

  1. Materials and Methods: Define the size of the study cohort. How many were recruited and what was the drop-out rate?

178 patients were recruited and all finished the study. This is now added in the manuscript.

  1. Line 131: What does this mean, ‘corrected for their species-specific response’?

This is now better described.

  1. Results: It would have been helpful to see data from a non-HCV control group.

It is not advisable to compare data of two different measurements because there are some variations between different measurements. Therefore we can not show data of any controls. Please apologize this limitation.

   What is the purpose of Figure 1B?

This was deleted.

  1. What is the relevance of diabetes to HCV? This is not clear. It should be introduced before the results section.

This is now described at the beginning of the Introduction.

  1. Figure 2 could be presented better. It is difficult to interpret along with Figure 6.

Figure 2 and figure 6 were replaced by a figure showing the different CE species as boxplots.

  1. Discussion: Again, a favourable and adverse CE profile is mentioned but at no point is this defined. What makes a CE profile ‘good’ or ’bad’?

This is now better explained in the Introduction.

Round 2

Reviewer 1 Report

Authors addressed raised issues appropriately.